# Robots Thinking Fast and Slow: On Dual Process Theory and Metacognition in Embodied AI

Ingmar Posner

Applied AI Lab, Oxford Robotics Institute

Oxford University, UK

Email: ingmar@robots.ox.ac.uk

*Abstract*—**Recent progress in AI technology has been breathtaking. However, many of the advances have played to the strengths of *virtual environments*: infinite training data is available, risk-free exploration is possible, acting is essentially free. In contrast, we require our robots to robustly operate in real-time, to learn from a limited amount of data, take mission- and sometimes safety-critical decisions and increasingly even display a knack for creative problem solving. To bridge this gap, here we offer an alternative view of recent advances in AI. In particular, we posit that, for the first time, roboticists can draw meaningful functional parallels between AI technology and components identified in the cognitive sciences as pivotal to robust operation in the real world: Dual Process Theory and metacognition. Revisiting recent work in robot learning, we establish the building blocks of a Dual Process Theory for *robots* and highlight potentially fruitful future research directions towards delivering robust, versatile and safe embodied AI.**

## I. INTRODUCTION

Recent advances in AI technology have built significant excitement as to what machines may be able to do for us in the future. Progress is truly inspirational. But where, you may ask, are the *robots*? Why can I buy a voice assistant but not a robust and versatile household robot? The answer lies in the fact that embodiment - the notion of a physical agent acting and interacting in the real world - poses a particular set of challenges. And opportunities.

Machines are now able to play Atari, Go and StarCraft. However, success here relies on the ability to learn cheaply, often within the confines of a virtual environment, by trial and error over as many episodes as required. This presents a significant challenge for embodied agents acting and interacting in the real world. Not only is there a cost (either monetary or in terms of execution time) associated with a particular trial, thus limiting the amount of training data obtainable, but there also exist safety constraints which make an exploration of the state space simply unrealistic: teaching a real robot to cross a real road by trial and error seems a far-fetched goal. What's more, embodied intelligence requires tight integration of perception, planning and control. The critical inter-dependence of these systems, coupled with limited hardware, traditionally leads to fragile performance and slow execution times.

Cognitive science suggests that, while humans are faced with similar complexity, there are a number of mechanisms which allow us to safely and efficiently act in the real world. One prominent example is *Dual Process Theory* popularised by Daniel Kahneman's book *Thinking Fast and Slow* [15]. Dual Process Theory postulates that human thought arises as a result of two interacting processes: an unconscious, involuntary – intuitive – response and a much more laboured, deliberate reasoning. Our ability to assess the quality of our own thinking – our capacity for *metacognition* – also plays a central role.

If we accept that Dual Process Theory plays a pivotal part in our own interactions with the world, the notion of exploring a similar approach for our robots is a tantalising prospect towards realising robust, versatile and safe embodied AI. If we can establish a meaningful technology equivalent - a Dual Process Theory for *robots* - mechanisms already discovered in the cognitive sciences may cast existing work in new light. They may provide useful pointers towards architecture components we are still missing in order to build more robust, versatile, interpretable and safe embodied agents. Similarly, the discovery of AI architectures which successfully deliver such dual process functionality may provide fruitful research directions in the cognitive sciences.

## II. A DUAL PROCESS THEORY FOR ROBOTS

While artificial intelligence research has drawn inspiration from the cognitive sciences from the very beginning, we posit that recent advances in machine learning have, for the first time, enabled meaningful parallels to be drawn between AI technology and components identified by Dual Process Theory. This requires mechanisms for machines to intuit, to reason and to introspect – potentially drawing on a variety of metacognitive processes.

### A. Machine Learning, Intuition and Reasoning

Machine learning is essentially an *associative* process in which a mapping is learned from a given input to a desired output (or intermediate representation) based on information supplied by an *oracle*. We use the term oracle here in its broadest possible sense to refer to both inductive biases and supervisory signals in general. In a brazen break with standard deep learning terminology, we also refer to an oracle's knowledge being *distilled* into a machine learning model. And we take as a defining characteristic of an oracle that it is in some sense resource intensive (e.g. computationally, financially, or in terms of effort or energy invested).

The learning of mappings of inputs to outputs has, of course, been a theme in ML for decades. However, in the context of Dual Process Theory, the advent of deep learning has afforded

our agents principally two things: (i) an ability to learn arbitrarily complex mappings; and (ii) an ability to execute these mappings in constant time. Together with an ability to learn structured, task-relevant embeddings even in an unsupervised manner, this affords researchers a different view on the computational architectures they employ: by learning ever more complex mappings from increasingly involved oracles we now routinely endow our agents with an ability to perform complex tasks at useful execution speeds. Direct human supervision, reinforcement learning, task demonstrations, complex learned models as well as the increasingly popular concept of system-level self-supervision – machines teaching machines – all fit into this narrative. Consequently, it is straight forward to cast recent results in AI and robotics into this paradigm. In game play, for example, DeepMind's AlphaGo Zero [29] as well as the closely related *Expert Iteration* algorithm [2] distil knowledge from Monte Carlo Tree Search (the oracle) and self-play into a model which predicts value and probability of next move given a particular board position. In robotics, OpenAI's Learning Dexterity project [1] distils knowledge from reinforcement learning using domain randomisation (the combined oracle) into a model which can control a Shadow Hand to achieve a certain target position in a dexterous manipulation task. In the context of autonomous driving, the authors of [3] distil, via the automatic generation of training data, hundreds of person-hours of systems engineering into a neural network model which predicts where a human might drive given a particular situation. Intuitive physics models predict the outcome of a particular scenario by a learner trained on data arrived at through physical simulation (see, for example, [30, 18, 31, 20, 11, 14]).

The key observation here is that, faced with an everyday challenge like game-play, driving or stacking objects, we do not tend to write down the governing laws of the process, nor do we use them explicitly to analyse the particular setup. Instead, we have a gut-feeling, an *intuitive* response. Importantly, owing to their ability to mimic the expertise of an oracle in a time (or generally resource) efficient manner, one might view the execution of a neural network model as analogous to an intuitive response. As already aptly noted by Kahneman [15]: intuition is recognition. And of course we also have access to a (very) broad class of oracles, which we might (generously perhaps, but with artistic license) refer to as reasoning systems. These then constitute direct analogues to System 1 and System 2. A Dual Process Theory for robots has thus firmly moved within reach.

### III. METACOGNITION IN A WORLD OF TWO SYSTEMS

The narrative of distilling knowledge into rapidly executable neural network models allows us to achieve significant, often game-changing, computational gains. However, as roboticists we are still faced with a substantive and foundational challenge when it comes to applying machine learning systems in the real world: the routine violation of the assumption that our systems face independent and identically distributed training and test data. In practice, together with the approximate nature of our algorithms, this leads to inference which is often overconfident and which can only loosely be bounded (if at all)

using traditional methods (e.g. [10, 27, 25]). It results in our robots lacking the ability to reliably know when they do not know and to take appropriate remedial action. Despite many attempts over the years at remedying this shortcoming (e.g. [28, 13, 19, 23, 7]) we are still no closer to a practicable solution. Yet a Dual Process Theory perspective may well provide a way forward.

Astonishingly, humans can be said to suffer from many of the same issues as machines. In particular, we are notoriously bad at knowing when we do not know – and, to the best of our knowledge, as yet no theoretical bounds are available. We operate in significantly non-stationary (in the statistical sense) environments. Yet – we do operate. Much of this success is commonly attributed to our *metacognitive* abilities [5, 9]: the process of making a decision, the ability to know whether we have enough information to make a decision and the ability to analyse the outcome of a decision once made.

One of the interesting aspects of a Dual Process Theory for robots is the fact that - given the analogy holds - metacognition finds a natural place in this construct: it bridges the two systems by regulating the intuitive, almost involuntary response of System 1 with a supervisory, more deliberate one of System 2. Do not trust your intuition, think about it. But only where appropriate, which is really the crux of the matter. Failure (or deliberate deception) of this mechanism is, of course, what gives rise to the cognitive biases now so well described in the literature [15]. Examining research on metacognition, therefore, might shed new light on how to tackle the *knowing-when-you-dont-know* challenge. And as an added bonus we now get machines with their own cognitive biases.

A direct implication of using machine learning models to provide intuitive, System 1 responses is that these models must assume operation within the data distribution encountered during training, whereas System 2 algorithms need to generalise beyond it. Robust, real-world performance thus seems to require computationally efficient policies empirically tuned to a particular task and environment *as well as* more computationally intensive approaches capable of systematic generalisation in that they are robust to variations in the task and environment. In humans, these two forms of processing interact in that System 2 can suppress, inform and even train System 1 responses [15]. While we can hypothesise that uncertainty plays a role in this information exchange there may be other mechanisms at work which could be exploited in robotics.

### A. Performance Prediction and a Feeling of Knowing

Ample evidence exists that humans represent and use estimates of uncertainty for neural computation in perception, learning and cognition [9]. However, how *metacognitive* uncertainties are derived and utilised is only gradually being discovered. Special, metacognitive circuitry in the human brain suggests knowledge integration above and beyond raw perceptual signals. Moreover, recent work on *multi-sensory* perception suggests that metacognition is also instrumental in discovering *causal* structures in order to form a coherent percept from multi-modal inputs [9].

In *Thinking Fast and Slow* [15] Kahneman exemplifies the responsibilities of System 1 and System 2 with a number of simple questions. For example, what is $2 + 2$? Or what is $2342114 \div 872$? The former elicits a System 1 response (a recall operation). The latter triggers the need for pen and paper - deliberate reasoning (System 2). One of the mechanisms regulating this routing – or algorithm selection – has been identified by metacognition researchers as the Feeling of Knowing Process [26]. It is executed near instantaneously and is able to make an (in the majority of cases) appropriate choice even based on only parts of the question. In the example above, by the time you have read "2+" your brain will have decided that you likely already know the answer to the question and only need to retrieve it. The *Feeling of Knowing* process therefore enables humans to effectively choose a cognitive strategy likely to succeed in a given circumstance: for example recall vs. reasoning – fast vs. slow.

Consider this in the context of robotics. And let us conjecture that the Feeling of Knowing Process is itself an intuitive (System 1) response. This immediately points at a set of now viable technical approaches in which, for example, the outcome of a downstream system (either in terms of success/failure or in terms of confidence in outcome) given a particular input is distilled into a machine learning model[1]. Such predictive models of performance are now relatively commonplace in the robotics literature. They have a long-standing track record in predicting task success in manipulation and complex planning tasks (e.g.[22, 16, 17, 24, 21]) and are increasingly used, for example, to predict the performance of perception and vision-based navigation systems (e.g. [12, 6, 8]).

Of course, distilling performance prediction into a machine learning model is but one way of giving a machine a Feeling of Knowing. We do not propose that it is the only - or even the best - way. Instead we highlight it as a functional equivalent to a recognised cognitive process, which is already being put to good use in robotics.

## IV. Conclusion and the Way Ahead

Our exposition so far makes the case that there is significant merit – perhaps now more than ever – in exploring constructs from Dual Process Theory and metacognition specifically for *robot* learning. In highlighting functional equivalents already established in robotics we aim to sketch an initial technical blueprint for components on either side of the systemic divide. As such, it is of course intended to stimulate discussion. The extent to which Dual Process Theory and metacognition can benefit robot learning and the mechanisms to best integrate them into our AI architectures are, as yet, unknown. However, one exciting aspect of a Dual Process Theory for robots is that there now exists a tantalising avenue within which to contextualise and along which to direct research.

When considering Dual Process Theory in robot learning, architecture designs that effectively get the best of both worlds remain an open challenge. Do both systems run in parallel? On demand? Is there an explicit handover between deliberate planning and low-level intuitive policies? Are there indeed two

[1]Statistical outlier detection also falls into this category.

separate systems or is it in fact a continuum of processes capable of fulfilling either part? Likewise, how should information be transferred between the systems and at what rate?

Perhaps unsurprisingly, the connection between reinforcement learning (RL) in particular and Dual Process Theory has not gone unnoticed. The authors of [4] highlight the role of slow learning particularly in the context of creating representations and inductive biases which later enable fast learning such as encountered in, for example, episodic RL and meta-RL. It is conceivable, therefore, that the fast and slow paradigm can be effected via a variety of inductive biases - both architectural (as advocated here in explicitly considering two separate systems) and learned. The view offered in [4] is therefore complementary to ours.

Finally, we note that opportunities exist in this context not only when it comes to improving robot learning. Cognitive science requires often complex experimental setups which, by design, need to disrupt the agent's learning process. Robotics, on the other hand, allows design and close inspection of the mechanisms involved. Therefore, as also noted in [4], the discovery of AI architectures which successfully deliver dual process functionality may also provide fruitful research directions towards advancing the state of the art in the cognitive sciences.

## Acknowledgments

This paper is in large parts based on a blog post written by the author in June 2019 reflecting many conversations that were had over the past decade both in the Oxford Applied AI Lab and beyond. The perspective offered here therefore has been influenced significantly by students and collaborators past and present. More recently, Robots Thinking Fast and Slow has also been the subject of discussion at the 2019 Montreal Robotics and Machine Learning Summit sponsored by Element AI and co-organised by the author, who is indebted to the Summit participants for their views in helping to shape parts of this narrative.

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
