# OpenReview forum: "Robots Thinking Fast and Slow: On Dual Process Theory and Metacognition in Embodied AI"
_roboticsfoundation.org/RSS/2020/Workshop/RobRetro — RobRetro 2020_

### Official Review · AnonReviewer1 · 2020-06-21
**Great retrospective on connections between cognitive science and robot learning**

**Rating:** 9
**Confidence:** 4

**Review:**

This retrospective draws a connection between recent trends in robot learning, specifically the application of deep learning techniques to robotics, and concepts such as Dual Process Theory and Metacognition from cognitive science. Drawing these parallels can help transfer insights from either field to the other, facilitating further progress in real world robotics and better understanding of human cognition. I really like the ideas presented in this paper and further formalisation of these ideas would be very interesting to the broader community.

A few points:
1. The article is well written and puts forth several examples to elucidate the presented concepts. Particularly, I liked the analogy likening the execution of a neural network to an intuitive low-level response, and the potential for regulating such responses via metacognition through a higher level, deliberative response.
2. While the article views modern learning approaches through the lens of dual process theory, it is interesting to note that such ideas have driven the architecture of many of our current robotics systems as they use a bi-level architecture, a high-level planner that is deliberative and a low-level controller that is instantaneous or reactive. Recent work in RL and robot learning has also started to look at such hierarchical structures for task decomposition, skill learning etc. It would be useful to draw connections to these areas of work.
3. Looking at things from the point of hierarchical learning several related questions arise. Should the depth of these systems be limited to a bi-level structure or do we go towards a multi-level architecture? In what way does information flow in these hierarchies, is it always limited to top-down deliberation and corrections or do we have layers of interconnection that can reason better? A formalisation of the current presentation could delve deeper into such issues.
4. The article also highlights at a key question that needs to be tackled in many of our current systems: “Knowing when we know, and knowing when we do not know”. As the article points out, it would be great if we can borrow tools from cognitive science to better inform the structuring of our learning systems.

Overall, this is a nice article that draws several interesting analogies between cognitive science and robot learning which will be of interest to the broader robotics community.

---

### Official Review · AnonReviewer2 · 2020-06-22
**Reasonably good perspective paper, but could be clearer and more comprehensive**

**Rating:** 7
**Confidence:** 4

**Review:**

Overall, I think the main thesis of the paper is reasonable (at least the part about dual processes).  Much of current AI research essentially involves training from data that was more costly to obtain (a slow/costly process generates data that is distilled into a system which can use it).  I think the paper equivocates a little around whether current ML uses slow processes that are actually similar to what a human would do.  The section in which the notion of the oracle is introduced could be a little more explicit.  I think the author is basically stating that the oracle could be any slow/costly process.  This ambiguity about the oracle feels significant, because it would seem to me to matter a lot whether the oracle is something like a human (or set of humans) or another slower computation performed by the AI system itself. So specifically, in trying to make this framework comprehensive for contemporary ML/AI ("Direct human supervision, reinforcement learning, task demonstrations, complex learned models as well as the increasingly popular concept of system level self-supervision – machines teaching machines – all fit into this narrative"), it needs to be more clear whether the claim is broadly that ML is about distillation from slow processes, or if, more narrowly, an aim for AI systems should be to mimic human-style cognition by consisting of these two systems (slow and fast).  In short, I feel as though this perspective is a bit inconsistent as to which of these two claims it is making (maybe it is both, but it isn't sufficiently clear which point is being made at which time).

Aside from the above point about the core thesis, I think the support for the claim is generally reasonably good.  However, there are a few points where addition detail could be explored.  Again, on the point about supervised training, it could and perhaps should be emphasized that human labeling is the slow process, and possibly this should be factored into the proposed framework more explicitly, especially given the ethical considerations that arise from human labeling (both in terms of biases and in terms of labor considerations).  When discussing driving, one example is given, but driving is explored at an industry scale already and e.g. Tesla openly discusses supervised training from actual drivers (talks by Karpathy).  A question these cases raise is whether an overall AI system involving humans as the "slow" system is really one system (following up on my request for disambiguation above).

In terms of coverage, I also feel there is an omission of work in the "motor control" and robotics space.  I think more research than is mentioned has seriously explored the themes discussed by this paper.  There are actually a number of examples that aim to build general/controllable skill modules, specifically through supervised distillation of trajectories that were more costly to acquire.

Two papers by Igor Mordatch and colleagues, really do a nice job of distilling from planners.
"Combining the benefits of function approximation and trajectory optimization." (2014)
"Interactive control of diverse complex characters with neural networks." (2015)

Researchers from DeepMind have done work with distilling behaviors into reusable skill modules:
"Robust imitation of diverse behaviors" (2017)
"Neural probabilistic motor primitives for humanoid control" (2019)

And another group learns policies from teleoperation in a simulated robotic environment:
"Learning latent plans from play" (2019)

Arguably approaches such as "guided policy search" (2013) also have an element of slow and fast systems, though perhaps this one is a matter of perspective.

I think the above work, and likely other papers from the robotics and motor control literature, already provide some concrete effort and direction along the lines of what is argued for in this paper.  I would encourage discussing some combination of the references above and perhaps other work in this topic area.

Finally, in the section on metacognition, it almost feels like another use of slow reasoning is introduced, or is metacognition just separate?  I thought, up to this point, that the slow system was the one which generated the target data and the fast system was the one into which the targets was distilled.  In this section, I find it a little unclear what is really being stated in the last three paragraphs of section III (before A).  While metacognition seems important, I don't feel like the current paper communicates explicitly enough the relationship between metacognition and the fast and slow processes.  Upon searching back to the introduction to help clarify, I simply found the statement "Our ability to assess the quality of our own thinking – our capacity for metacognition – also plays a central role."  Ultimately, it feels like the dual process ideas and the assertions related to metacognition are separate points that have been stitched together?  I think if these are simply separate issues, it should be clearer why the paper focuses on them, and what if any relationship they have.

PS: After reading AnonReviewer1's review, I believe my point about "motor control" literature is likely quite related to AnonReviewer1's point 2.

---

### Decision · Program_Chairs · 2020-06-25

Accept